# Genome-Wide Identification of the PAL Gene Family in *Camellia nitidissima* and Functional Characterization of *CnPAL1* Gene by In Vitro Expression

**DOI:** 10.3390/genes16111251

**Published:** 2025-10-23

**Authors:** Hexia Liu, Bo Li

**Affiliations:** 1College of Smart Agriculture, Yulin Normal University, Yulin 537000, China; liuhexia2010@163.com; 2Guangxi Key Laboratory of Smart Agriculture for Special Horticultural Crops, Yulin Normal University, Yulin 537000, China

**Keywords:** *Camellia nitidissima*, *PAL* gene family, expression analysis, in vitro enzyme activity analysis

## Abstract

**Background**: *PAL* genes are crucial for plant growth and stress response, yet studies on the *PAL* gene family in *Camellia nitidissima* are sparse. **Methods**: The *PAL* gene family was screened from the entire genome of *C. nitidissima*, and their physicochemical properties, chromosomal locations, intraspecific and interspecific collinearity, conserved motifs, phylogenetic trees, cis-acting elements, and gene structures were analyzed. The expression patterns of the *CnPAL* genes were compared across different tissues, and the highly expressed *CnPAL1* gene was expressed in prokaryotes, and its enzyme activity was validated using UPLC-MS technology. **Results**: The results revealed that six *CnPALs* were identified in the *C. nitidissima* genome, distributed unevenly across six chromosomes. The CnPAL proteins shared similar physicochemical properties, with highly conserved motifs and gene structures. Promoter analysis showed multiple cis-acting elements in the *CnPALs* genes. Intra-species collinearity analysis revealed that all *CnPALs* were collinear with multiple *PAL* genes in *C. nitidissima*, while inter-species collinearity analysis indicated that *CnPALs* were collinear with the *PAL* genes in *Camellia oleifera* and *Camellia sinensis*. Furthermore, the transcriptomic data of *C. nitidissima* demonstrated tissue-specific expression of the *CnPALs*, although qRT-PCR validation showed some discrepancies with the sequencing result. The qRT-PCR revealed varied expression patterns among the six *CnPALs*, with the *CnPAL1* gene showing relatively higher expression levels. Subsequently, cloning, prokaryotic expression, and enzyme activity analysis confirmed the effective catalytic activity of the CnPAL1 protein. **Conclusions**: This study lays the foundation for understanding the functions of *CnPAL* genes and offers insights for genetic improvement of *C. nitidissima*.

## 1. Introduction

*Camellia nitidissima*, part of the Camellia genus in the Theaceae family, is primarily distributed in Guangxi, China [1]. This plant is rich in bioactive compounds, including tea polyphenols, polysaccharides, flavonoids, saponins, amino acids, and trace elements [2,3]. Studies have demonstrated that *C. nitidissima* exhibits various physiological activities, including hypoglycemic, lipid-lowering, antioxidant, and immunomodulatory effects [4,5]. Traditionally, it has been used to treat conditions like hypertension, dysentery, pharyngitis, and hematochezia [6]. In Guangxi, *C. nitidissima* is popular both as a health beverage and a medicinal herb in traditional Chinese medicine. As a food-medicine plant, its flowers and leaves are particularly high in flavonoids, with the flowers containing the most [7,8]. These flavonoids not only serve as fundamental pigments responsible for flower coloration [9,10] but also play crucial roles in regulating physiological activities and enhancing stress resistance. They also contribute to the medicinal properties of *C. nitidissima*, including anti-cancer, antioxidant, and lipid-lowering effects [4,5,11].

Flavonoids are crucial plant secondary metabolites that perform diverse physiological functions, such as protecting against UV and intense light, aiding seed dispersal by attracting predators, defending against pathogens, and contributing to flower pigmentation [12]. Recent studies also highlight their potential in preventing and treating chronic human diseases, such as cancer, cardiovascular disorders, diabetes, inflammation, obesity, and aging [13,14]. Structurally, flavonoids are divided into seven main subclasses: anthocyanins, proanthocyanidins, flavonols, flavones, flavanones, isoflavones, and chalcones [15], making them the largest group of natural phenolic compounds. The core of the flavonoid biosynthetic pathway is highly conserved and branches from the phenylalanine pathway, which originates from the shikimate pathway. Phenylalanine ammonia-lyase (PAL) is a key rate-limiting enzyme that catalyzes the conversion of phenylalanine to cinnamic acid, thus directing carbon flux from primary to secondary metabolism [16,17]. Importantly, studies have identified *PAL* genes as positive regulators of flavonoid accumulation, as seen in *AtPAL1* and *AtPAL2* in *Arabidopsis thaliana* [18] and *EfPAL1* and *EfPAL2* in *Euryale ferox* [19].

*PAL* genes in plants typically form multigene families, with substantial variation in family size among species. For instance, *A. thaliana* harbors four *PAL* genes [20], while *Ginkgo biloba* contains 11 [21]. Other species exhibit varying numbers, including four in *Nicotiana tabacum* [22], 14 in *Solanum tuberosum* [23], and 15 in *Vitis vinifera* [24]. The development of high-throughput sequencing technologies and the availability of reference genomes for numerous plant species have facilitated the identification of the *PAL* gene families in various organisms, such as *A. thaliana* [25], *Brassica napus* [26], and *Solanum lycopersicum* [27]. Furthermore, the expression levels of *PAL* gene family members are dynamically regulated by environmental stimuli and hormonal signals [28]. Given this variability, systematic identification of the *PAL* gene family members and analysis of their expression patterns are essential for understanding their biological functions across different species.

Previous studies have shown a positive correlation between *PAL* gene expression and flavonoid content in *C. nitidissima* [29], suggesting its potential role in flavonoid accumulation. However, the regulatory mechanism of *CnPALs* in flavonoid biosynthesis is still unclear, and the *PAL* gene family in *C. nitidissima* has not been characterized. To address the gap, this study employs bioinformatics to systematically identify the *PAL* gene family in the chromosome-level genome of *C. nitidissima* and analyze their physicochemical properties, chromosomal locations, intraspecific and interspecific collinearity, conserved motifs, phylogenetic trees, cis-acting elements, and gene structures. Subsequently, the tissue-specific expression patterns of the *CnPAL* genes are investigated using transcriptomic data and validated via quantitative real-time PCR (qRT-PCR). The highly expressed *CnPAL1* gene is chosen for further functional characterization, including cloning, prokaryotic expression, and enzyme activity analysis in vitro using UPLC-MS technology. This research aims to provide a theoretical foundation for elucidating the functions of the *CnPAL* gene family and to support future breeding efforts in *C. nitidissima*.

## 2. Materials and Methods

### 2.1. Materials

*C. nitidissima* plants were cultivated at the nursery of Yulin Normal University, Guangxi, China (22°40′58″ N, 110°11′27″ E). The nursery is situated in a temperate climate zone, with an average annual temperature of approximately 22 °C and abundant rainfall averaging 1650 mm per year. The soil is classified as red laterite. *C. nitidissima* is grown under a broad-leaved plantation with about 50% canopy cover. The plants, 8 years old, exhibit vigorous growth and are currently in their prime flowering phase. Various tissues, including roots, stems, leaves, flower buds, and fully bloomed flowers, were collected in triplicate (Figure 1). Immediately after collection, the samples were flash-frozen in liquid nitrogen and stored at −70 °C in an ultra-low temperature freezer for later analysis.

### 2.2. Identification and Physicochemical Characterization of the CnPAL Genes in C. nitidissima

The *C. nitidissima* genome and GFF annotation files (GWHFILD00000000.1) have been downloaded from the National Genomics Data Center (NGDC) [30]. Amino acid sequences of phenylalanine ammonia-lyase (PAL) proteins retrieved from the Arabidopsis Information Resource (TAIR) (https://www.arabidopsis.org/) and aligned using BLASTp (Nucleotide-Nucleotide BLAST 2.12.0+). Hidden Markov Model (HMM) files corresponding to the Lyase_aromatic domain (PF00221), a characteristic PAL domain, were downloaded from the Pfam database (http://pfam.xfam.org, accessed on 20 May 2025) [31]. Using HMMER 3.0 (http://hmmer.org/ accessed on 21 May 2025), the *C. nitidissima* genome was scanned to identify the *PAL* genes containing the PAL domain, after which incomplete and redundant sequences were removed. To ensure accuracy, the presence of the PAL domains was confirmed using Pfam, SMART (http://smart.embl-heidelberg.de/ accessed on 21 May 2025), and InterPro (http://www.ebi.ac.uk/interpro/ accessed on 21 May 2025).

The ProtParam tool (https://web.expasy.org/protparam/ accessed on 21 May 2025) [32] was used to calculate the physicochemical characteristics of CnPAL proteins. These properties include length of amino acid sequence, theoretical isoelectric point (pI), molecular weight (Mw), instability index, aliphatic index, and grand average of hydropathicity (GRAVY). For predicting subcellular localization of the CnPAL proteins, the Wolfpsort online platform (https://wolfpsort.hgc.jp/ accessed on 21 May 2025) was utilized, with the highest-confidence prediction chosen as the final result.

### 2.3. The Analysis of Evolutionary and Structural Characterization of the CnPAL Genes

PAL amino acid sequences from *C. nitidissima*, *A. thaliana*, and *Camellia sinensis* were aligned using ClustalW with default settings. Phylogenetic trees were constructed via Maximum Likelihood (ML) in MEGA12 [33], with 1000 bootstrap replicates. The best-fit substitution model for sequence evolution was identified as JTT+F+R6 using ModelFinder. Then, IQ-TREE was used to perform phylogenetic tree with default settings, and the tree was visualized using iTOL (https://itol.embl.de/ accessed on 21 May 2025). Conserved motifs in *C. nitidissima* PAL proteins were identified using MEME Suite (http://www.omicsclass.com/article/67 accessed on 21 May 2025), with parameters set to a maximum of 10 motifs and a width of 6–50 amino acids [34]. Gene structures (CDS/UTR) were extracted from the file of genome annotations and visualized with TBtools v2.225 [35].

### 2.4. Chromosomal Location, Gene Duplication, and Collinearity Analysis of the CnPAL Genes

The ‘Gene Location Visualize’ module in TBtools v2.225 [35] was used to map the *CnPAL* genes on chromosomes. Homologous *PAL* genes in *C. nitidissima* were identified via BLASTp using criteria of identity > 90 and E-value < 10^−10^. Intra-specific collinearity analysis was performed using MCScanX [36] with default parameters. For inter-specific collinearity analysis, genome sequences of *C. sinensis* (ShuzaochaV2) and *Camellia oleifera* (GCA_025200525.1) were downloaded from TPIA and NCBI, respectively. The MCScanX program [36] analyzed collinearity among the three species with default parameters. Results were visualized using the modules of Advanced Circos, Comparative Genomics and OneStepMCScanX-SuperFast in TBtools [35].

### 2.5. Analysis of Cis-Acting Elements in the CnPAL Genes Promoter

The promoter sequence for each *CnPAL* gene was identified as the 2000 bp region upstream of the start codon (ATG). Promoter sequences were extracted using TBtools [35], and cis-regulatory elements were predicted using the PlantCARE database (https://bioinformatics.psb.ugent.be/webtools/plantcare/html/ accessed on 21 May 2025) [37]. The visualization of these cis-acting elements was performed using the Basic Biosequence View and HeatMap modules in TBtools [35].

### 2.6. Analysis of Expression Levels of the CnPAL Genes

The sequencing data of *C. nitidissima* comprises 36 transcriptome datasets, including 12 different tissues with three replicates per tissue. These datasets were obtained from NCBI (accession: PRJNA392895). Raw sequencing data was processed using fastp [38] to eliminate low-quality reads, yielding clean reads for downstream analysis. HISAT2 [39] was employed to align paired-end reads to the reference genome. StringTie [40] reconstructed transcripts from alignment results, while RSEM [41] calculated Fragments Per Kilobase of transcript per Million mapped reads (FPKM) values for the *CnPAL* genes across samples. The transcriptome expression data of the *PAL* genes was converted to log2 and then normalized row-wise using the Z-score method. Subsequently, hierarchical clustering was performed on the rows. Expression patterns were visualized using the HeatMap module in TBtools [35].

### 2.7. qRT-PCR Validation of the CnPALs

Specific primers for six *CnPAL* genes were designed using Primer Premier 5.0 (Appendix A), and their specificity was confirmed by qRT-PCR. Total RNA extracted from roots, stems, leaves, flower buds, and fully open flowers was reverse-transcribed into cDNA using an Invitrogen reverse transcription kit (Carlsbad, CA, USA). qRT-PCR reactions employed SYBR Green Dye (TianGen Bio-Chem Technology, Beijing, China) with 18S rRNA as the internal reference gene. Each sample underwent triplicate analysis, and relative expression levels were calculated using the 2^−∆∆CT^ method [42].

### 2.8. Construction of Expression Vectors for the CnPAL1 Gene

PCR products were subjected to electrophoresis to identify the target band, which was then extracted using the kit (FastPure Gel DNA Extraction Mini Kit V21.1) (Novogene, Nanjing, China). An expression plasmid was constructed using the LR Cloning Enzyme (Invitrogen, Carlsbad, CA, USA). The ligation mixture (5 µL total volume) contained: 0.5 µL pDEST15 plasmid vector, 0.5 µL target fragment, 0.5 µL LR Cloning Enzyme, and 3.5 µL ddH2O. The ligation mixture was incubated at 25 °C for 6 h, before being transformed into DH5α competent cells and cultured overnight. Monoclonal colonies were then selected for PCR verification.

### 2.9. Prokaryotic Expression and Purification of CnPAL1 Protein

The constructed expression vector plasmid (pDEST15-CnPAL1) was transformed into competent BL21 (DE3) cells and incubated overnight at 37 °C. Selected monoclonal colonies were then inoculated into 3 mL of LB medium containing ampicillin and shaken at 180 rpm overnight at 37 °C. This culture was diluted 1:100 into 250 mL of LB medium with ampicillin and incubated under the same conditions until OD600 reached 0.6. Induction was initiated by adding 0.2 mM IPTG, followed by incubation at 20 °C with shaking at 180 rpm for 18 h. A 1 mL sample of the uninduced culture was kept as a control (CK). Induced cells were harvested by centrifugation at 4 °C and 8000 rpm for 20 min. The resulting pellet was resuspended in 30 mL of 50 mM Tris-HCl buffer (300 mM NaCl, pH 7.8) and disrupted by high-pressure homogenization with 1 mM PMSF. The lysate was centrifuged at 4 °C and 12,000× *g* rpm for 1 h to obtain the supernatant. CnPAL1 protein was purified using a GST-tagged purification kit (Yisheng, Shanghai, China), with uninduced lysate serving as a control. Expression and purification were confirmed by 10% SDS-PAGE analysis.

### 2.10. Assessment of Recombinant Enzyme Activity

The experimental group, consisting of purified CnPAL1 protein, and the negative control, featuring heat-inactivated CnPAL1, were separately added to reaction mixtures with substrate and buffer. These mixtures were then incubated at 30 °C for 1 h. The enzymatic assay was performed in a total volume of 100 µL, comprising 5 µL of 1 M Tris-HCl (pH 8.0), 15 µL of 100 mM L-phenylalanine (L-Phe), 2 µL of 100 mM DTT, 5 µg of purified CnPAL1 protein, and nuclease-free ddH2O to reach the desired volume. The reaction products were quantified using ultra-performance liquid chromatography coupled with tandem mass spectrometry (UPLC-MS/MS).

LC separation was conducted on an Agilent SB-C18 column (1.8 µm, 2.1 × 100 mm) using a gradient of ultra-pure water (Phase A) and 5–95% acetonitrile (Phase B), both containing 0.1% formic acid. For the first 9 min, acetonitrile was maintained at 95% for 1 min. From 10.00 to 11.10 min, the acetonitrile concentration decreased to 5% and was held at 5% until the 14 min mark. The flow rate was set at 0.35 mL/min at 40 °C, with 2 µL injections.

MRM analysis was performed on a triple quadrupole mass spectrometer equipped with electrospray ionization (ESI) at 500 °C with an ion spray voltage of ±5.5/4.5 kV. The gas pressures were set at 50/60/25 psi for GS1, GS2, CUR, respectively, and nitrogen served as the collision gas at medium intensity. Declustering potential (DP) and collision energy (CE) were individually optimized for each ion pair. MRM ion pairs were monitored based on the metabolites eluted during each period.

### 2.11. Statistical Analysis

All experiments were performed using a completely randomized design with three replicates for each treatment. Data analysis was conducted using one-way ANOVA, followed by Tukey′s honest significant difference test for multiple comparisons (*p* < 0.05) utilizing SPSS version 16.0.

## 3. Results

### 3.1. Genome-Wide Analysis of the PAL Gene Family in C. nitidissima and Physicochemical Characterization of PAL Proteins

Genome-wide sequence analysis identified six *PAL* gene family members in *C. nitidissima*, named CnPAL1 to CnPAL6 based on their chromosomal locations. The coding sequences (CDSs) of these genes ranged from 2112 to 2184 base pairs, encoding proteins with 703–727 amino acids and molecular weights between 76.65 and 77.76 kDa. The aliphatic indices of these proteins, from 90.29 to 95.74 (Table 1), suggest potential thermostability. Subcellular localization predictions revealed distinct compartmentalization patterns: CnPAL1-CnPAL4 are likely in chloroplasts, while CnPAL5 and CnPAL6 are targeted to the plasma membrane. The theoretical isoelectric points analysis indicated all six PAL proteins have acidic properties, with isoelectric points between 5.84 and 6.29. Hydrophilicity analysis demonstrated negative grand average hydropathicity values for all CnPAL proteins, confirming their hydrophilic nature. Notably, the instability indices of all six proteins were below 40, classifying them as stable proteins.

### 3.2. Gene Structure and Conserved Motif Analysis of the CnPAL Gene Family

The genetic structural analysis of the *CnPAL* gene family revealed that all *CnPAL* genes have the same genetic structure (Figure 2), consisting of two exons separated by one intron in each gene. Protein functional analysis of the CnPAL amino acid sequences, using NCBI’s Conserved Domain Search tool, identified a highly conserved PLN02457 (phenylalanine ammonia-lyase) domain common to the entire family. This domain is part of the Lyase_I_like superfamily. Furthermore, conserved motif analysis of the CnPAL protein sequences found ten conserved motifs present in all six CnPAL proteins. These structural and functional similarities highlight the significant conservation among *CnPAL* family members.

### 3.3. Phylogenetic Analysis of the CnPAL Gene Family

This study constructed a phylogenetic tree featuring 17 PAL proteins from *A. thaliana*, *C. sinensis*, and *C. nitidissima* (Figure 3). The analysis classified the PAL family proteins into four distinct subfamilies (I, II, III, and IV). Notably, the four PAL proteins from *A. thaliana* were primarily distributed in Clade I and Clade II. In contrast, CnPAL proteins (CnPAL1 and CnPAL3) clustered with two tea PAL proteins (CsPAL2 and CsPAL6) in Clade III. Meanwhile, CnPAL2, CnPAL4, CnPAL5, and CnPAL6 grouped with the remaining five tea PAL proteins in Clade IV. The phylogenetic analysis indicated that CnPAL proteins have closer evolutionary relationships with tea PAL proteins from the same genus and family than with those from *A. thaliana*.

### 3.4. Analysis of Promoter Cis-Acting Elements of the CnPAL Gene Family

Sequences spanning 2000 bp upstream of the six *CnPAL* genes were extracted for promoter cis-acting element analysis. The analysis identified 32 distinct regulatory elements within the *CnPAL* gene family, categorized into four main classes: light-responsive, hormone-responsive, stress-responsive, and plant growth metabolism-responsive elements. Each class was further divided into more functionally specific subtypes, totaling 14 subtypes in the *CnPAL* gene promoters (Figure 4a). Quantitative analysis revealed hormone-responsive elements (78 elements) and stress-responsive elements (76 elements) were the most prevalent, while plant growth metabolism-related elements were the least common (21 elements). Among these, MYC (stress-responsive) and ABRE/ERE (growth metabolism-responsive) elements showed the broadest distribution, present in all *C. nitidissima PAL* gene promoters. In contrast, nine elements, including MBS, WUN-motif, CAT-box, GCNA_motif, A-box, CCGTCC-motif, CCGTCC-box, ACE, and I-box, appeared only in specific *PAL* gene family members. Variation in promoter cis-acting elements was evident, with *CnPAL3* containing the highest element count (48) and *CnPAL5* the lowest (28) (Figure 4b). This cis-element distribution pattern suggests that *CnPAL* genes collectively play roles in hormone signaling, stress adaptation, and developmental processes in *C. nitidissima*.

### 3.5. Chromosomal Location and Collinear Analysis of the CnPAL Gene Family

Chromosomal mapping revealed that six *CnPAL* genes are located on six distinct putative chromosomes (Chr01, Chr02, Chr07, Chr10, Chr11, and Chr12), with each chromosome containing one *PAL* gene (Figure 5). Intra-specific collinearity analysis demonstrated that all *CnPAL* genes have collinear relationships with multiple genes, leading to the identification of 14 gene duplication events among them. These events include *CnPAL1* with *CnPAL2*, *CnPAL3 CnPAL4*, and *CnPAL5*; *CnPAL2* with *CnPAL3*, *CnPAL4 CnPAL5*, and *CnPAL6*; *CnPAL3* with *CnPAL4*, *CnPAL5*, and *CnPAL6*; *CnPAL4* with *CnPAL5* and *CnPAL6*. All these duplication events were identified as segmental duplications (Figure 6). Inter-species collinearity analysis of *PAL* genes among *C. nitidissima*, *C. oleifera*, and *C. sinensis* revealed that six *CnPAL* genes are collinear with the *PAL* gene families in both related species, with 18 and 27 gene pairs identified in *C. oleifera* and *C. sinensis*, respectively (Figure 7). This pattern indicates conserved evolutionary relationships among the *PAL* genes within the Camellia genus.

### 3.6. Expression Analysis of the CnPAL Gene Family

To elucidate the characteristics and functions of the *CnPAL* gene family, we analyzed the expression profiles of six *CnPAL* genes using transcriptomic sequencing data across nine tissue types (roots, leaves, buds, calyces, petals, stamens, pistils, fruits) and five flowering stages (Figure 8). The expression patterns were classified into three distinct types. The first pattern showed enrichment in reproductive organs. For instance, *CnPAL2* and *CnPAL3* had similar profiles, with predominant expression in reproductive organs and during flowering stages, suggesting potential roles in flavonoid biosynthesis in *C. nitidissima*. The second pattern exhibited low expression across all tissues, with *CnPAL5* consistently showing low levels. *CnPAL6* and *CnPAL4* were preferentially expressed in leaves, fruits, and young flower buds, while *CnPAL1* displayed root-specific expression, fitting the third pattern of tissue-specific expression. qRT-PCR validation in five tissues (roots, stems, leaves, flower buds, and open flowers) revealed some discrepancies with the transcriptomic data (Figure 9). Notably, *CnPAL1* showed the highest expression, especially in stems and leaves. *CnPAL2*, *CnPAL3*, and *CnPAL5* exhibited moderate expression in nutrient-accumulating organs, while *CnPAL6* was predominantly expressed in root. This multi-layered expression analysis offers comprehensive insights into the functional diversification of the *CnPAL* genes in *C. nitidissima*.

### 3.7. Cloning, Prokaryotic Expression, and Enzyme Activity Analysis of CnPAL1 Gene

The *CnPAL1* gene from *C. nitidissima* was successfully cloned via reverse transcription-polymerase chain reaction (RT-PCR) (Appendix A). This cloned gene was then inserted into a prokaryotic expression vector (pDEST-17) and heterologously expressed in *Escherichia coli* BL21 cells. The recombinant protein was subsequently purified (Appendix A). Enzymatic activity assays involved incubating the purified CnPAL1 protein (experimental group) and heat-inactivated CnPAL1 protein (control group) with reaction substrates and buffers at 30 °C for 1 h. UPLC-MS/MS analysis demonstrated that the experimental group produced novel reaction products. This confirmed that the CnPAL1 protein from *C. nitidissima* exhibited catalytic activity, efficiently converting L-phenylalanine to trans-cinnamic acid (Figure 10).

## 4. Discussion

### 4.1. Characteristics of the CnPAL Gene Family in C. nitidissima

The phenylpropanoid metabolic pathway constitutes one of the three principal secondary metabolic pathways in plants. It begins with phenylalanine as the starting substrate and leads to the production of various secondary metabolites, including flavonoids, phenolic compounds, and lignin. These metabolites are indispensable for plant defense against biotic and abiotic stresses [43]. Phenylalanine ammonia-lyase (PAL) acts as a key enzyme linking primary metabolism to the phenylpropanoid pathway, playing a significant role in plant metabolic networks [44].

In this study, six *PAL* gene family members were identified in the 3.0 Gb genome of *C. nitidissima* [30], distributed across six distinct chromosomal loci. The number of genes in *C. nitidissima* exhibits evolutionary similarities with species like *C. sinensis* (7 genes) [45], *Epimedium pubescens* (7) [46], *Citrus grandis* (5) [47], and *Citrus clementina* (5) [47]. The physicochemical properties of the CnPAL proteins, such as amino acid sequence length, molecular weight, isoelectric point, instability index, and grand average of hydropathicity, demonstrate limited variability. This is consistent with observations for PAL proteins in *C. sinensis* [45], *Cucumis sativus* [48], and *Arachis hypogaea* [49]. Comparative subcellular localization analysis reveals that CnPAL proteins in *C. nitidissima* have similar distribution patterns to the *PAL* gene family in *Brassica oleracea* [50]. Specifically, CnPAL1-CnPAL4 are localized in chloroplasts, suggesting their involvement in chloroplast-related processes, while CnPAL5 and CnPAL6 are cytoplasmic proteins potentially mediating functions within the cytoplasm. Gene structure analysis indicates that all *CnPAL* genes possess a conserved architecture comprising two exons and one intron, and ten conserved motifs are uniformly distributed across the family. This structural conservation is also seen in closely related species such as *C. sinensis*, where *PAL* genes exhibit nearly identical exon-intron organization and motif composition [45,51]. Phylogenetic reconstruction places the six *CnPAL* genes predominantly in Clades III and IV, with collinear relationships between the *CnPAL* genes of *C. nitidissima* and their orthologs in tea plants. These findings collectively suggest that the *CnPAL* genes and tea *PAL* genes originate from a common ancestral gene, and *PAL* genes within the same phylogenetic subfamily likely maintain functionally conserved roles.

### 4.2. Analysis of the Expression Characteristics of the CnPAL Genes and Functional Validation of the CnPAL1 Gene

During plant growth and development, *PAL* gene expression is tissue-specific. For instance, *Artemisia annua*’s *PAL* genes are most active in leaves [52], similar to *Picrorhiza kurrooa* [53], whereas *Juglans regia*’s *PAL* genes are mainly expressed in roots [54]. In this study, the expression patterns of the *CnPAL* genes in *C. nitidissima* were analyzed using both transcriptomic sequencing and qRT-PCR. The finding showed that *CnPAL* genes also exhibited tissue-specific expression. Transcriptomic data indicated *CnPAL1* was highly expressed in roots, *CnPAL6* and *CnPAL4* were elevated in leaves, fruits, and young flower buds, and *CnPAL2* and *CnPAL3* were preferentially expressed in reproductive organs. However, qRT-PCR results differed, showing *CnPAL1* was highly expressed in stems, leaves, flower buds, and mature flowers. *CnPAL2*, *CnPAL3*, *CnPAL5*, and *CnPAL6* were primarily found in vegetative organs, while *CnPAL4* exhibited relatively low expression in blooming flowers. The discrepancies likely arise from batch effects, due to non-identical sample sets. These findings suggest functional redundancy and complex regulatory mechanisms governing the *CnPAL* gene expression in *C. nitidissima*.

The flavonoid composition of *C. nitidissima* is primarily composed of flavonoids and proanthocyanidins [7,55]. These secondary metabolites are biosynthesized through pathways regulated by transcription factors, including MYB, bHLH, and WD40 proteins [56,57]. In this study, analysis of the promoter cis-acting elements of the *CnPAL* gene family reveals that *CnPAL* promoters contain one MBS element bound by MYB, 24 MYC elements and 17 G-box elements bound by bHLH transcription factors. Since the bHLH-responsive cis-elements exhibit a striking enrichment in *C. nitidissima* and the critical role of cis-acting elements in gene function [58], it is inferred that *CnPAL* genes are primarily regulated by bHLH transcription factors, affecting *PAL* gene expression and function. Previous studies demonstrated that *PAL* gene expression is positively correlated with flavonoid content [59]. Consequently, the highly expressed *CnPAL1* gene validated by qRT-PCR was cloned and expressed in prokaryotes, and its enzymatic activity was confirmed in vitro. The results demonstrated that *CnPAL1* effectively catalyzes the conversion of L-phenylalanine into trans-cinnamic acid. However, further transgenic experiments are required to determine whether the CnPAL1 protein enhances flavonoid accumulation in *C. nitidissima*.

## 5. Conclusions

This research identified six *CnPAL* genes with complete domains from the *C. nitidissima* genome. Analysis revealed that CnPAL proteins exhibit acidic, stable, and hydrophilic properties. Phylogenetic analysis demonstrated that *CnPAL* genes in *C. nitidissima* are more closely related to those in tea plants (*C. sinensis*). Promoter analysis identified multiple cis-acting elements responsive to bHLH transcription factors, suggesting potential bHLH-mediated transcriptional regulation. The *CnPAL* gene family displayed tissue-specific expression patterns. Prokaryotic expression and in vitro enzymatic assays confirmed that the highly expressed *CnPAL1* gene catalyzes the conversion of L-phenylalanine to trans-cinnamic acid. These findings provide critical insights for further understanding the characteristics and exploring the biological functions of the *CnPAL* gene family.

## Figures and Tables

**Figure 1 genes-16-01251-f001:**
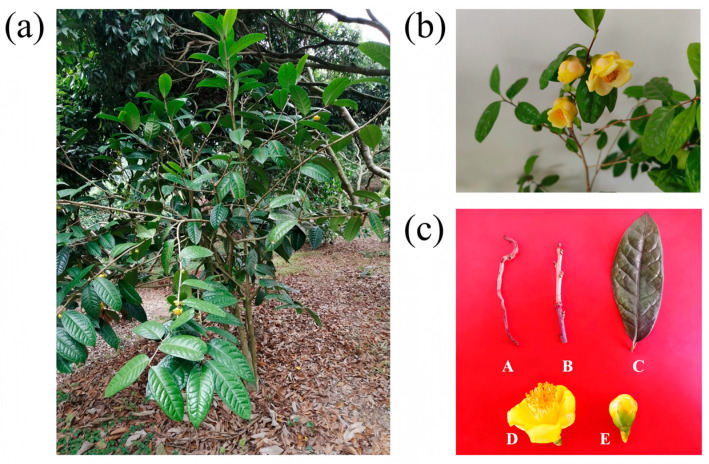
Different tissues of *C. nitidissima*. (**a**) Growth conditions of *C*. *nitidissima* in the nursery. (**b**) Blooming morphology of *C*. *nitidissima* flowers. (**c**) A, B, C, D and E represent root, stem, leaf, blooming flower, and bud of *C. nitidissima*, respectively.

**Figure 2 genes-16-01251-f002:**
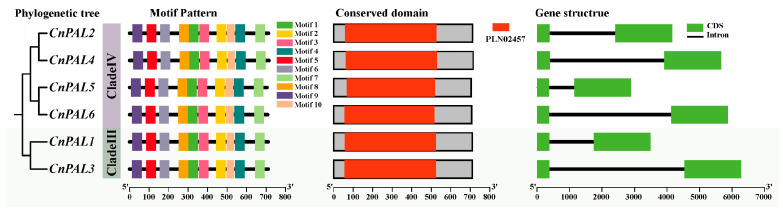
Illustrates the evolutionary relationships, conserved motifs, protein domains, and gene structures of the *CnPAL* gene family. In the depiction of protein conserved structural domains, gray boxes denote non-conserved domains, while red boxes signify PAL protein conserved domains. In the gene structure of the *CnPALs*, green boxes represent exons, and black lines indicate introns.

**Figure 3 genes-16-01251-f003:**
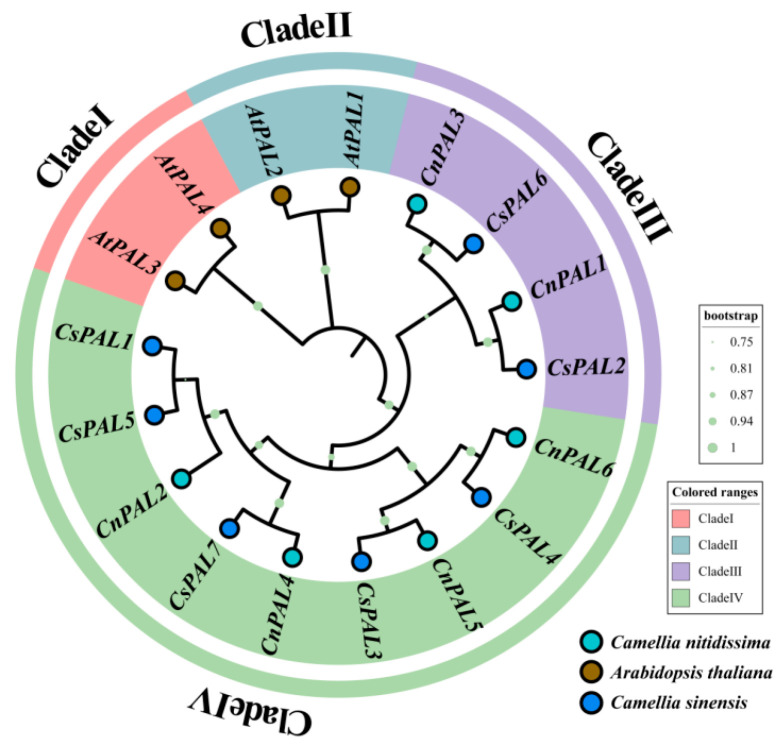
Phylogenetic tree of the PAL family proteins in *C. nitidissima*, *A. thaliana*, and *C. sinensis*. Phylogenetic trees were constructed via Maximum Likelihood (ML) in MEGA12 with 1000 bootstrap replications.

**Figure 4 genes-16-01251-f004:**
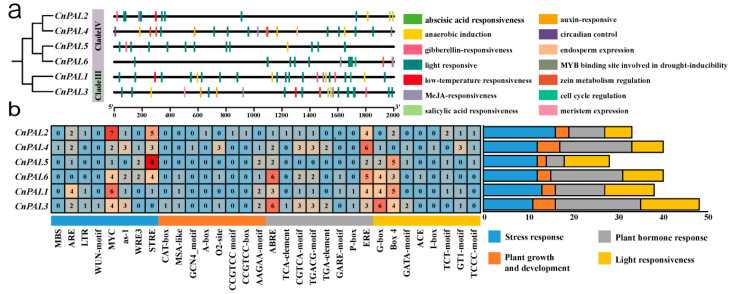
Analysis of cis-acting element in the promoter region of *CnPAL* genes in *C. nitidissima*. (**a**) Types and distribution of cis-acting element in the *CnPALs* promoter. (**b**) Statistical analysis of various cis-acting elements types in the *CnPALs*. Each box displays the number of cis-acting elements, with color intensity indicating the number of promoters.

**Figure 5 genes-16-01251-f005:**
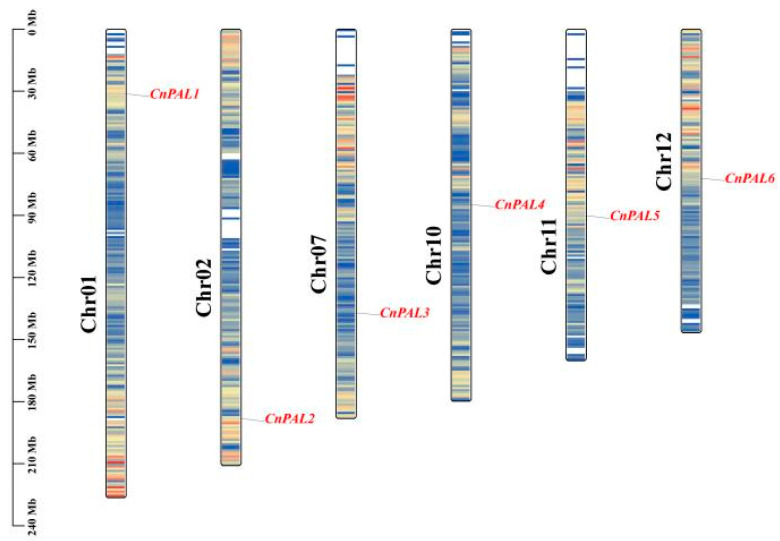
Chromosomal localization map of the *PAL* gene family in *C. nitidissima*. The elongated structure depicts chromosomes, with chromosome numbers shown in black on the left. The names of the *PLA* genes are highlighted in red font on the right side. The scale bar on the left indicates the chromosome length in millions of bases (Mb).

**Figure 6 genes-16-01251-f006:**
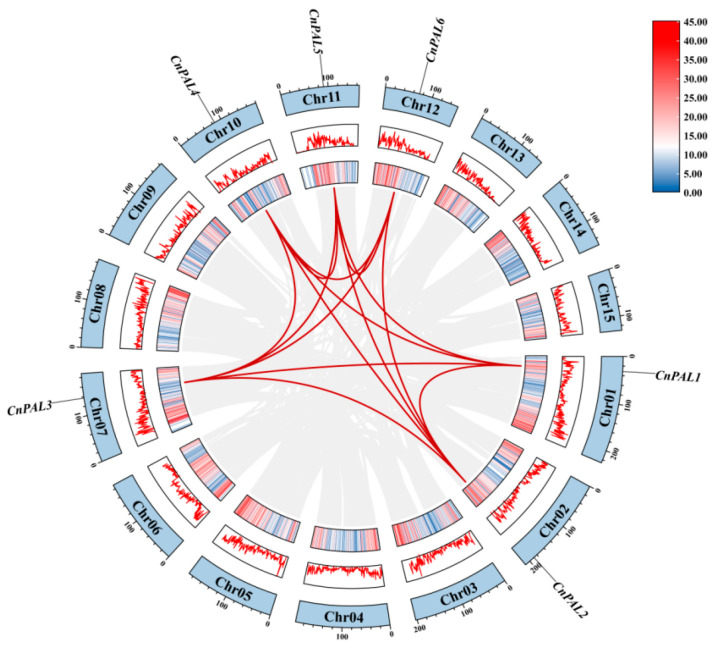
Intraspecific collinearity analysis of the *CnPALs*. The outer circular track represents 15 chromosomal-level scaffolds (Chr01-Chr15), while the inner circular track indicates gene density ranging from 0 to 45. The red lines represent collinear gene pairs of the *CnPALs*.

**Figure 7 genes-16-01251-f007:**
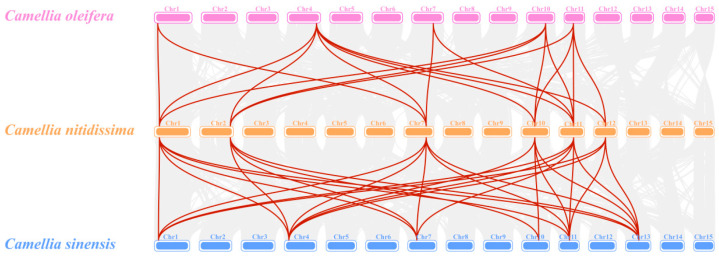
Interspecific collinearity analysis of the *PAL* genes among *C. nitidissima, C*. *sinensis* and *C*. *oleifera*. Gray lines indicate collinear blocks between the genomes of *C. nitidissima* and *C. sinensis*, as well as *C. nitidissima* and *C. oleifera*. Red lines represent collinearity of *PAL* gene pairs between the genomes of *C. nitidissima* with *C. sinensis* and *C. nitidissima* with *C. oleifera*.

**Figure 8 genes-16-01251-f008:**
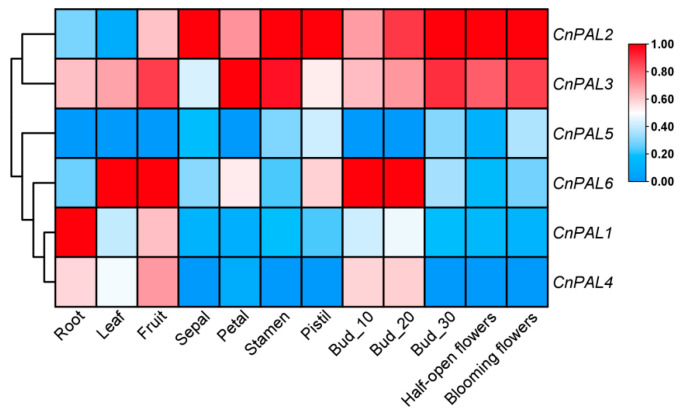
Heatmap showing the expression levels of the *CnPAL* genes in different tissues. The data have been standardized with the Z-score method, and row-wise clustering has been applied. Bud_10, bud_20, and bud_30 refer to flower buds measuring 10, 20, and 30 mm in length, respectively.

**Figure 9 genes-16-01251-f009:**
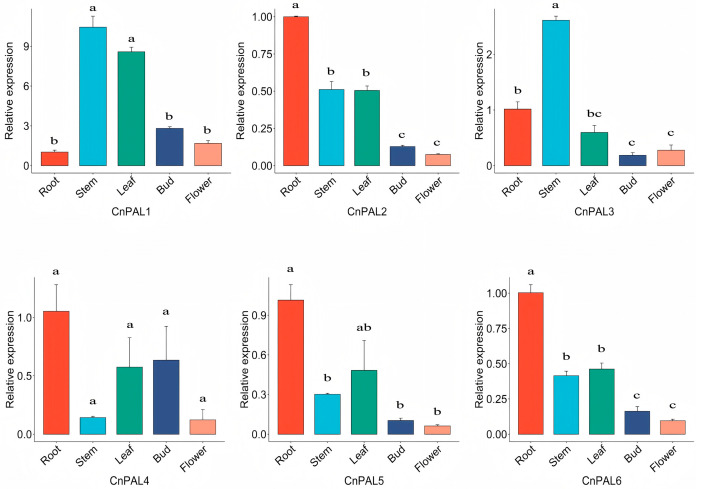
Relative expressions of *CnPALs* in different cultures by qRT-PCR. Data are mean $\bar{x}$ ± *SE* from three independent biological replicates. Different letters indicate significant differences at *p* < 0.05 by one-way ANOVA test.

**Figure 10 genes-16-01251-f010:**
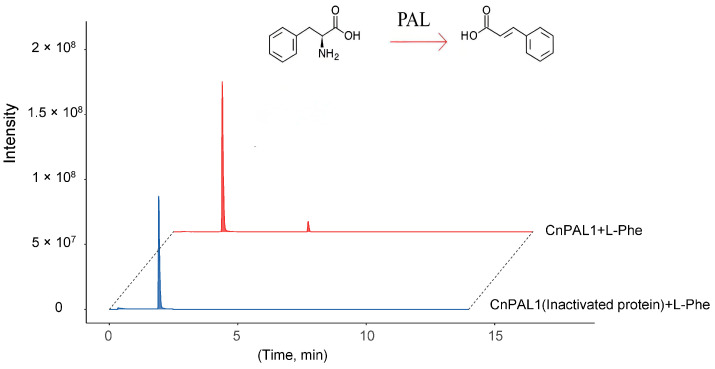
UPLC-MS/MS analysis of enzyme reaction products from the incubation of CnPAL1 infusion protein. The blue line represents the negative control with inactivated protein of CnPAL1 (CK), while the red line shows the experimental group with purified CnPAL1 protein. The theoretical retention time (RT) of the reactant, L-phenylalanine, is 1.97 min, and the retention time for the reaction product, trans-cinnamic acid, is 5.29 min.

**Table 1 genes-16-01251-t001:** Physical and chemical properties of the *PAL* gene family in *C. nitidissima*.

Gene Name	Gene ID	Number of Amino Acids	Molecular Weight	Theoretical pI	Instability Index	Aliphatic Index	Grand Average of Hydropathicity	Subcellular Localization Prediction
*CnPAL1*	Cpet01g03790.t1	709	77,358.56	5.90	32.88	95.74	−0.11	Chloroplast
*CnPAL2*	Cpet02g23820.t1	711	77,454.46	6.29	33.96	90.55	−0.19	Chloroplast
*CnPAL3*	Cpet07g18270.t1	709	77,304.45	6.29	34.81	92.19	−0.15	Chloroplast
*CnPAL4*	Cpet10g09570.t1	714	77,760.93	5.97	36.73	90.29	−0.15	Chloroplast
*CnPAL5*	Cpet11g11740.t1	703	76,646.69	5.84	35.26	93.93	−0.13	Cytoplasmic
*CnPAL6*	Cpet12g14570.t1	706	77,084.20	6.16	33.81	91.74	−0.15	Cytoplasmic

## Data Availability

The data have been deposited to the National Center for Biotechnology Information (NCBI) under accession number PX254426.

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
