# Peer review of "Genome-Wide Identification of the PAL Gene Family in Camellia nitidissima and Functional Characterization of CnPAL1 Gene by In Vitro Expression"

_genes, 2025, doi:10.3390/genes16111251_

Round 1
Reviewer 1 Report
Comments and Suggestions for Authors
The paper is interesting and provides new insights into a gene family in an interesting crop. But there are some points that require improvement.
Figure 1 is of low quality. Please, make closer and individual pictures of each organ, as they would be of great help for readers unfamiliar with this system.
Figures 2-8: Please include in the figure legend the software that you have used and the parameters.
Figure 9: Please, include a statistical analysis of the data to check the level of significance of the observed changes.
Comments on the Quality of English Language
The quality of English is very low. It is difficult to read, as some grammar constructions used by the authors are quite strange. Please, make a thorough revision. I advise the use of professional proofreading.
Author Response
To reviewer #1:
Comment 1: Figure 1 is of low quality. Please, make closer and individual pictures of each organ, as they would be of great help for readers unfamiliar with this system.
Answer: According to the associate editor and the comments of reviewers, we have made extensive modifications to our figures (Fig. 1), in page 3, line 106-109. I hope that the changes I have made resolve all your concerns about the article.
Comment 2: Figures 2-8: Please include in the figure legend the software that you have used and the parameters.
Answer: According to the associate editor and the comments of reviewers, we have made extensive modifications to our figures. For example, Figue. 2, in page 7, line 259-263; Figue. 3, in page 7, line 276-278; Figue. 4, in page 8, line 299-302; Figue. 5, in page 8, line 319-322; Figue. 6, in page 9, line 325-327; Figue. 7, in page 9, line 330-334; Figue. 8, in page 10, line 356-359. Besides, additional drawing parameters have been supplemented in the method section.
I hope that the changes I have made resolve all your concerns about the article.
Comment 3: Figure 9: Please, include a statistical analysis of the data to check the level of significance of the observed changes.
Answer: According to the associate editor and the comments of reviewers, we have made extensive modifications to our figures. For example, Figue. 9, in page 10, line 360-363. Besides, additional drawing parameters have been supplemented in the method section, in page 5, line 225-229. I hope that the changes I have made resolve all your concerns about the article.
Comment 4: The quality of English is very low. It is difficult to read, as some grammar constructions used by the authors are quite strange. Please, make a thorough revision. I advise the use of professional proofreading.
Answer: Thanks for your very thoughtful suggestion. We have engaged native English-speaking experts to refine and polish the paper.
Reviewer 2 Report
Comments and Suggestions for Authors
Review report
The manuscript presents a comprehensive analysis of the PAL gene family in Camellia nitidissima, a species of both ornamental and medicinal importance. The authors performed genome-wide identification, structural and phylogenetic characterization, cis-regulatory element analysis, expression profiling, and functional validation of one PAL gene (CnPAL1). Overall, the study is well designed and addresses a relevant topic in plant molecular biology and secondary metabolism. The combination of bioinformatics, transcriptomic analysis, and enzymatic validation provides valuable insights into the PAL gene family in C. nitidissima. The manuscript is generally well-written, logically structured, and the findings are clearly presented. However, there are some methodological issues that need to be addressed to strengthen the manuscript before acceptance.
2.1 Materials
- The growth conditions (light intensity, photoperiod, temperature, humidity, soil type, fertilization) are not described. This limits reproducibility and may affect gene expression patterns.
- No information about the developmental stage or age of plants at sampling.
- Number of biological replicates collected per tissue is not mentioned.
2.4 Chromosomal Location and Collinearity Analysis
- The criteria for defining homologous genes (e.g., identity threshold, coverage) are not described.
- BLASTP E-value (“< 10^-10”) is given, but coverage cutoff missing.
2.6 Transcriptome Expression Analysis
- No mention of replicates, biological variation, or statistical testing for differential expression.
- Data validation (e.g., correlation between replicates, mapping rate) is missing.
2.7 qRT-PCR Validation
- Statistical analysis of qRT-PCR data (replicate number, standard deviation, p-value thresholds) not described.
- Statistical analysis methods are missing throughout — no mention of how variation or significance was evaluated.
Author Response
To reviewer #2:
Comment 1:
2.1 Materials
- The growth conditions (light intensity, photoperiod, temperature, humidity, soil type, fertilization) are not described. This limits reproducibility and may affect gene expression patterns.
- No information about the developmental stage or age of plants at sampling.
- Number of biological replicates collected per tissue is not mentioned.
Answer: Thanks very much for your suggestion.
According to the associate editor and the comments of reviewers, we have made extensive description of the growth conditions, the developmental stage or age of plants at sampling, and number of biological replicates in page 3, line 96-105. I hope that the changes I have made resolve all your concerns about the article.
Comment 2: 2.4 Chromosomal Location and Collinearity Analysis
- The criteria for defining homologous genes (e.g., identity threshold, coverage) are not described.
- BLASTP E-value ("< 10^-10”) is given, but coverage cutoff missing.
Answer: Thanks very much for your suggestion. Homologous genes were identified through BLASTp alignment with a sequence identity threshold of >90 % and E-value < 10-10. The relevant content has been added to the Methods section, in page 4, line 144-146. I hope that the changes I have made resolve all your concerns about the article.
Comment 3:
2.6 Transcriptome Expression Analysis
- No mention of replicates, biological variation, or statistical testing for differential expression.
- Data validation (e.g., correlation between replicates, mapping rate) is missing.
Answer: Thanks very much for your suggestion. The replicates, biological variation, or statistical testing for transcriptome expression analysis has been added in page 4, line 161-163 and line 168-171. I hope that the changes I have made resolve all your concerns about the article.
Comment 4:
2.7 qRT-PCR Validation
- Statistical analysis of qRT-PCR data (replicate number, standard deviation, p-value thresholds) not described.
- Statistical analysis methods are missing throughout — no mention of how variation or significance was evaluated.
Answer: Thanks very much for your suggestion. We have made extensive description of the statistical analysis of qRT-PCR data, in page 5, line 226-229 and page 10, line 359-363. And we hope the revised manuscript will be acceptable to you.
Once again, thank you very much for your constructive comments and suggestions which will help us both in English and in depth to improve the quality of the paper.